# Splenic Artery Pseudoaneurysms: The Role of ce-CT for Diagnosis and Treatment Planning

**DOI:** 10.3390/diagnostics12041012

**Published:** 2022-04-17

**Authors:** Fabio Corvino, Francesco Giurazza, Anna Maria Ierardi, Pierleone Lucatelli, Antonello Basile, Antonio Corvino, Raffaella Niola

**Affiliations:** 1Department of Vascular and Interventional Radiology, Cardarelli Hospital, 80131 Naples, Italy; francescogiurazza@hotmail.it (F.G.); raffaellaniola2@gmail.com (R.N.); 2Radiology Department, Fondazione IRCCS Cà Granda, Ospedale Maggiore Policlinico, 20122 Milan, Italy; amierardi@yahoo.it; 3Vascular and Interventional Radiology Unit, Department of Radiological, Oncological and Anatomo-Pathological Sciences, Sapienza University of Rome, 00185 Rome, Italy; pierleone.lucatelli@gmail.com; 4Radiology Unit 1, Department of Medical Surgical Sciences and Advanced Technologies “GF Ingrassia”, University Hospital “Policlinico-San Marco”, University of Catania, 95124 Catania, Italy; basile.antonello73@gmail.com; 5Motor Science and Wellness Department, University of Naples “Parthenope”, 80133 Naples, Italy; an.cor@hotmail.it

**Keywords:** splenic artery pseudoaneurysm, MDCTA, angiography, embolization

## Abstract

Splenic artery pseudoaneurysm (PSA) is a contained vascular wall lesion associated with a high mortality rate, generally related to pancreatitis, trauma, malignancy, iatrogenic injury, and segmental arterial mediolysis. Computed tomography angiography allows us to visualize the vascular anatomy, differentiate a PSA from an aneurysm, and provide adequate information for endovascular/surgical treatment. The present review reports on the main state-of-the-art splenic artery PSA diagnosis, differentiating between the pros and cons of the imaging methods and about the endovascular treatment.

## 1. Introduction

Pseudoaneurysms (PSAs) are vascular lesions generally due to a tear of the vessel wall contained in the adventitia of the artery or by the local hematoma surrounding PSA; unlike aneurysms, they are contained in all layers of the arterial wall [1]. Splenic artery PSAs could be due to pancreatitis, trauma, malignancy, or iatrogenic injury [2]. Another cause of PSAs formation—even if rarer—is segmental arterial mediolysis (SAM); SAM is a non-atherosclerotic, non-inflammatory vascular disease of unknown origin that could involve the visceral arteries of the abdomen, as well as the splenic artery [3].

Visceral artery PSAs are rare; however, because of more widespread and aggressive hepato-biliary surgery and percutaneous interventions, the incidence of visceral PSAs seems to have increased [4].

Moreover, in recent years the increase of availability of cross-sectional abdominal imaging, such as Computed Tomography Angiography (CTA) and Magnetic Resonance Imaging (MRI), has led to a higher detection rate of visceral aneurysms and PSAs. In about 40% of cases the location is the splenic artery [5].

PSAs, as they lack complete vascular wall structure, gradually increase in size and are under growing strain with the continuous impact of blood flow, resulting in sudden or massive bleeding. Indeed, the most common complication of PSA is a rupture, and the subsequent haemorrhage is correlated with a high risk of mortality, reaching rates of over 90%. A rapid diagnosis and an early treatment results in improved survival with a significant drop-in mortality rate [6].

Cross-sectional imaging, such as CT angiography study and/or angiography study, is essential for definite diagnosis and for providing information for endovascular/surgical treatment planning [7,8].

Endovascular treatment, such as embolization, has been emerging as a first-line therapy in splenic PSA management of various aetiologies related to its high technical success rate with a low morbidity rate compared to surgical treatment, especially in acute settings [9,10]. Even if technological advancement has led to an increase of early diagnosis, to date, the diagnosis of PSAs continues to be missed by a rate higher than 50% [11].

In this article, we review the general characteristics of main splenic artery PSAs, describing the techniques and findings of CT angiography studies, comparing it with other imaging techniques and, finally, some notes about endovascular treatment.

## 2. Etiology

PSAs are contained vessel wall lesions surrounded by hematoma itself, adjoining tissue, and/or fibrous reaction. The real frequency of splenic artery PSAs is underestimated because in literature there are mainly retrospective analysis and because the patients were asymptomatic in the course of the disease. Moreover, with the development of endovascular techniques it is very rare to have a histopathological proof of the diagnosis of pseudoaneurysm. From the limited case reports and case series available, the main causes of splenic artery PSAs are pancreatitis (in about 50% of cases), blunt or penetrating trauma (29%), iatrogenic injuries (7%), and, to a much lesser extent, direct neoplasm invasion by pancreatic or gastric cancer (2%) and SAM (less than 1%). PSAs vary widely in size with a range of 0.3–17 cm (mean, 4.8 cm) [10,11,12].

Acute pancreatitis PSAs are due to autodigestion and weakening of the walls of the nearby vessels by proteolytic pancreatic enzymes. Pancreatic enzymes could erode the vessel wall, causing necrotizing arteritis with complete destruction of the vessel architecture, dissolution of its elastic tissue and incipient vessel wall fragmentation (Figure 1). Moreover, hemorrhage due to PSAs bleeding could involve adjacent structures such as organs (stomach, colon), pancreatic duct, peritoneum, retroperitoneum, or pseudocyst if associated. Another pathophysiological explanation of PSA formation in pancreatitis is due to the spread of pancreatic enzymes related to severe inflammation: a pseudocyst could erode an adjacent vessel wall by the content of its pancreatic enzymes, by direct compression and subsequent vessel wall ischemia, inducing PSA formation, or by converting itself in a large PSA [13,14]. The true incidence of visceral arteries PSAs in pancreatitis is unknown; a wide range of 1.3–10% has been reported in pancreatitis patients, without difference between acute and chronic pancreatitis [15]. However, the splenic artery is the most common artery involved in about 35–40% of cases followed by the gastro-duodenal artery (20–25% of cases). Moreover, any visceral artery may be involved in PSA formation; PSAs related pancreatitis may develop in several additional visceral arteries, including pancreaticoduodenal, gastroepiploic, dorsal pancreatic, gastric, hepatic and superior mesenteric arteries [16]. Moreover, the literature also describes splenic artery PSA formation due to penetrating the gastric ulcer; likewise for pancreatitis, gastric enzymes caused necrosis and weakening of a vessel wall, which resulted in PSA formation. Peptic ulcer disease is a rare cause of splenic artery PSAs; only four such cases have ever been reported [17].

With the advancement of complex surgical operations, iatrogenic PSAs are relatively common encounters in clinical practice. The overall incidence of PSAs following abdominal and pelvic surgery is relatively low; however, the incidence of PSAs formation increases with the more invasive surgeries [18]. Vessel wall disruption may be related to a direct injury during abdominal procedure or surgery, various endoscopic treatments, or an indirect injury from post-operative perivascular inflammation or infection, resulting in the formation of PSA [19]. The process of clearing the lymph nodes around the artery with electrocautery and ultrasonic scalpel and removing vessel wall adventitia in addition to clamping of vessels may lead to vessel tear and subsequent contained vascular lesion development, such as PSAs [20,21] (Figure 2).

Abdominal trauma as a cause of splenic PSA is more often intrasplenic rather than isolated to the main splenic artery. Post-traumatic splenic artery PSAs are thought to occur as a result of shearing forces, causing damage to the arterial wall in cases of blunt trauma; rapid deceleration results in damage to the intima and elastic lamina of the vessel wall (Figure 3). A direct interruption of the vessel wall’s integrity is related to PSAs formation in penetrating trauma [22,23]. A very curious case report describes massive upper gastrointestinal bleeding, due to splenic PSA formation in communication with a coarse stomach ulcer, related to a previous endoscopic removal of a metal wrench in a patient suffering from Pica, the psychiatric disease [24]. PSAs development is usually related to blunt rather than penetrating trauma; Tessier et al. performed a review of the literature in 2003, which identified 157 total reported cases of pseudoaneurysms, 29% of which were attributable to blunt trauma [5].

Pancreatic cancer often invades adjacent organs and results in symptoms such as obstructive jaundice, duodenal obstruction, weight loss, and pain; vascular involvement and PSAs formation are very rare and generally secondary to acute pancreatitis related to pancreatic retention cysts with dilatation of the upstream pancreatic duct [25,26]. A more recent case report demonstrates the development of splenic artery PSA due to direct cancer invasion during chemotherapy in a patient with pancreatic cancer [27]. Moreover, a recent case report describes massive upper gastrointestinal bleeding secondary to a visceral communication between the patient’s advanced gastric malignancy and splenic artery PSA [28].

Finally, SAM was firstly described in 1976 by Slavin and Gonzalez-Vitale. It is a very rare disease, just over 100 cases of SAM have since been reported in the literature; the vessel most often associated with SAM is splenic artery, accounting for 28% of all SAM involvement [29,30]. SAM affects the outer layer of media muscle fibers, leading to smooth muscle cell vacuolar degeneration (Figure 4). The disruption of these vacuoles associated with reduction of its fluid contents results in a tear of the outer medial muscle from adventitia, with intramural hemorrhage and fibrin deposit. Vessel wall tears may be filled by fibrin, thrombi, or granulation tissue and it could lead to saccular aneurysm, PSAs, or complete thrombosis. SAM spared intima from these lytic changes where there is only minimal inflammation [11,12,13,14,15,16,17,18,19,20,21,22,23,24,25,26,27,28,29,30,31].

## 3. Symptoms

Early diagnosis and swift treatment are critical, as PSAs present the potential for a massive hemorrhage. A prompt diagnosis requires awareness of presenting symptoms. The main risk is their tendency to rupture, which it is a true medical emergency, and it may present as a hemorrhagic shock with massive hematemesis and diffuse abdominal pain. PSAs are almost always symptomatic (more than 75% of cases), mainly presenting as abdominal pain, acute gastrointestinal bleeding, and abdominal distension [13,14,15,16]. Acute GI hemorrhage evidenced by physical examination findings of melena and hematemesis is often attributed to direct bleeding into the gastrointestinal tract or into the pancreatic duct, a phenomenon known as “hemosuccus pancreaticus”. Those patients with extensive retroperitoneal hematoma, but without bleeding into the gastrointestinal tract, may present with pain only [20,21,22,23,24,25,26,27,28,29,30,31,32].

In patients with necrotizing pancreatitis or following pancreatic surgery, the onset of fresh bleeding from a drain, percutaneous, or surgical, is a threatening sign suggestive of intraabdominal bleeding, known as a “sentinel bleed”. In up to 90% of cases, initial minor bleeding could be followed by a massive blood loss; Sato et al. described sentinel bleeding in all 10 patients of their study as a predictive sign of massive bleeding following pancreatectomy [33]. On the other hand, some authors demonstrate that identification of a sentinel bleed in patients subject to pancreaticoduodenectomy was not associated with postoperative 30 days mortality, but about 60% of cases with secondary hemorrhage had a sentinel bleed and all of them had sepsis or a pancreatic leak [34]. The size of the splenic artery PSAs is not a determining factor in the risk of rupture [6].

Finally, a minor rate of patients (about more than 20%) are asymptomatic, and generally PSA diagnosis is typically incidental [10].

## 4. Diagnosis

Splenic artery PSAs are a clinical emergency; a prompt diagnosis could allow a specific life-saving treatment. Since there is no complete wall structure, PSAs could increase in size under the high flow arterial pressure and eventually result in a breakout or sudden bleeding. The risk of spontaneous rupture of a PSA is very high and sentinel bleeding is a prodromal symptom of massive hemorrhage [6]. Therefore, a rapid and accurate diagnosis is mandatory. The increasingly wider use of cross-sectional abdominal imaging, such as CTA and MRI, has led to a higher detection rate of visceral aneurysm in recent years [10,11,12,13,14,15,16,17,18,19,20,21,22,23,24,25,26,27,28,29,30,31,32,33,34,35]. However, PSAs may be considered underdiagnosed, and the diagnosis is confirmed only when complications occur [5].

Several imaging modalities, including plain radiographs, sonography, CT, and angiography, can be used to diagnose PSAs, mostly performed in cases of mildly symptomatic patients. X-rays are very rare helpful, especially in splenic artery aneurysms, showing prevertebral calcification or a calcified spot localized towards the splenic hilum [36,37].

### 4.1. Ultrasound

Ultrasound (US) baseline could be used to study splenic artery PSAs. However, US has poor results in diagnosis of splenic artery PSAs and could be useful especially in intraparenchymal or perihilar PSAs; indeed, overlying bowel gas, obesity, arteriosclerosis, and poor patient compliance considerably reduce its diagnostic sensitivity [38,39]. Moreover, due to its sensitivity to detect splenic artery aneurysms, and then also PSAs, <3 cm is poor [40].

On US studies using B-mode imaging, PSAs typically presented as cystic-like lesion (black hole sign), an ovoid or round structure comprised in a larger hyperechoic hematoma. The cystic lesion may have appeared to be transected by a fluid level, with a superficial anechoic aspect, and more deep echogenic material layering that expands itself during each systolic arterial pulsation; moreover, the more echogenic fluid appeared to be swirling and pulsating with direct arterial communication. The high velocity flow across the neck may cause a vibration across the adjacent soft tissue, inducing a speckling artefact on the color Doppler study [41].

The color Doppler study shows a swirling motion blood flow inside the lesion with a systolic flow in the pseudoaneurysm direction and diastolic flow in the opposite direction, which is filled with color, often with a “yin and yang” waveform pattern [42].

The pulsed-Doppler study shows on the aneurysm neck or in the sac close to it a “to and fro” bidirectional type waveform, with direct flow into the aneurysm sac during systole and reversed flow from the aneurysm sac to the parent arterial lumen during diastole. The “to and fro” flow pattern is synchronous with the cardiac cycle. This waveform may be dampened in the periphery of larger PA sacs, where it may be confused with that of a venous waveform [43].

Contrast enhanced ultrasound (CEUS) helps in the case of a suspected PSA lesion without a typical color flow signal; it could increase US study diagnostic accuracy. CEUS allows to demonstrate significant enhancement of a cystic/cystic-like region adjacent splenic artery and it could also to clearly display thrombus as a well-defined area without contrast enhancement [44].

When PSAs are not visible directly, US may be useful to demonstrate abdomen free fluid; this indirect sign could change subsequent treatment due to the clinical condition of a patient; if the hemodynamic is unstable, refer the patient to an explorative laparotomy, and if stable, to a CTA study [45].

### 4.2. Computed Tomography Angiography (CTA)

Over the past few years, the development of CT technologies, with faster improved scanners and modern post-processing software has meant that the sensitivity and specificity of CT angiography is approaching and superseding that of digital subtraction angiography (DSA) in many applications. To date, CTA is the most commonly used and sensitive technique to diagnose splenic artery PSAs; the sensitivity and specificity of CTA for the detection of arterial complications related to pancreatitis were 94% and 90%, respectively [10,11,12,13,14,15,16,17,18,19,20,21,22,23,24,25,26,27,28,29,30,31,32,33,34,35,36,37,38,39,40,41,42,43,44,45,46].

Moreover, in addition to diagnosis, CTA has been established as gold-standard for the treatment planning and follow-up of most disease of the abdominal arteries, including aorta and visceral arteries. [47]. The execution technique provides a multiphase scan (baseline, arterial/venous/delayed phases) that allow a panoramic assessment and necessary data for endovascular treatment of splenic artery PSAs. Intravenous administration is essential to obtain an adequate imaging; peripheral intravenous access with a large cannula, typically 18-G or 20 G in average-sized adults, and use of a power injector to maintain a high flow rate, are standard. Generally, arterial enhancement, crucial for the diagnosis, is provided by the intravenous administration of 70–120 mL of non-ionic iodinated contrast material, at a variable injection rate of 3–5 mL/s. Moreover, the acquisition delay time is individually determined by a bolus-tracking system or even by using an empirical delay, which usually ranges from 20 to 35 s [48,49]. The arterial phase is crucial not only for the anatomical definition of the vascular lesion, but also to prove and locate possible contrast extravasation. Splenic artery PSAs appear as a focal contrast-filled vessel wall outpouching bubble-shaped (saccular) budding from the lateral wall of artery generally rounded by a spontaneous hyperdense lesion in basal scans (local thrombus or hematoma). In contrast to active haemorrhage, PSAs maintain their shape on delayed phase imaging, whereas the contrast increases and changes shape in the setting of an active haemorrhage. Occasionally, the outpouching may not entirely fill with contrast, owing to the presence of thrombus [38,39,40,41,42,43,44,45,46,47,48,49,50]. In the case of a complicated pseudocyst, CTA study shows a focal region of enhancement into it, highly suspicious of PSA, after contrast administration. An exposed arterial segment surrounded by necrotic tissue is prone to the development of a pseudoaneurysm and bleeding; only CTA may demonstrate an exposed arterial segment, because DSA can study the vessel lumen and when extraluminal necrosis is present, the lumen may appear angiographically normal [51].

In the case of SAM, the most common arterial imaging findings are dissection (86% of cases), followed by aneurysm, beading or webs, occlusion and rind or wall thickening. Dissection may be seen as a longitudinal linear hypoattenuating intimo-medial flap when both true and false lumen are patent, or may present as a focal filling defect in case one of the lumens is thrombosed. Vessel contour irregularity with luminal narrowing may be present in the setting of superimposed thrombus [30,31,32,33,34,35,36,37,38,39,40,41,42,43,44,45,46,47,48,49,50,51,52]. Without pathologic disease confirmation, the diagnosis can only be described as an arterial dissection lacking a precise identity [53].

CTA, thanks to multiplanar reconstruction and 3D images, could provide essential information for pre-procedural planning influencing therapeutic choice (endovascular treatment or surgery) directly [54]. In our experience, thick maximum intensity projection images in all planes are very helpful for making the diagnosis of splenic PSAs.

A recent retrospective series analyzed the diagnostic accuracy of CTA in visceral artery PSAs diagnosis, and showed that in only 42% of cases PSAs were diagnosed correctly, with a more frequent involvement of the hepatic artery followed by the splenic one. Moreover, this paper shows the causes for missing diagnosis of visceral artery PSAs, summarizing it in four different reasons: missing of contrast media phase (in 36% of cases); artifact masking the PSA (in 20% of cases); overlooked PSA not recognized by the attending radiologist (in 42% of cases), and false interpretation of the CT imaging findings (in 6% of cases). In this series, about 42% of the missed PSAs were overlooked by a diagnostic radiologist generally due to alteration in their normal vessel anatomy subsequent to foregoing abdominal surgery. Since clinical and laboratory findings may be unspecific, an early and correct radiological diagnosis will most likely improve the outcome of patients with PSAs; including an arterial contrast media phase to the protocol CT, using techniques for metal artifact reduction and increased knowledge of the normal postoperative anatomy seems to be the most effective tips to reduce PSAs misdiagnosis rate [11].

### 4.3. Magnetic Resonance Imaging (MRI)

MRI plays a minor role in the diagnosis of splenic artery PSAs, as it is usually not the reference imaging modality to evaluate patients with suspected visceral PSAs. However, splenic PSA diagnosis may be incidental in the evaluation for other disease; it could be an adjunct to US studies or an alternative to CTA, especially in patients with contraindication to contrast media or with reduced renal function (the MRI contrast agent is less nephrotoxic compared to CT) [55,56].

T1- and T2-weighted acquisition demonstrate blood products or thrombus within or along the PSA wall; at the same time, the patent portion of a PSA mimic the signal intensity of the other arteries. MR angiography T1-weighted images with intravenous gadolinium show, as CTA, PSA as a focal, round enhancing outpunching budding from the adjacent wall of artery with an intensity signal similar to adjacent patent arteries. The MRI angiography study could be obtained even without a contrast agent, with the use of specific angiographic sequences (time of flight and flow sensitive sequences). Up-to-date MRI angiography allows us to characterize vascular flow thanks to high contrast resolution [57].

The main disadvantages of MRI are the long acquisition times and the need of patient compliance, making it unsuitable in an emergency setting. MRI could have a major role in the follow-up of treated patients with Time-Resolved MRA; however, even if MRI is less susceptible to a metallic artifact compared to CTA, metallic (endovascular embolic agent after percutaneous treatment) and bowel artifact could make post-embolization evaluation very hard [58,59].

## 5. DSA and Treatment

Independent of their associated symptoms or diameter, pseudoaneurysm should always be treated. The following factors may affect the clinician’s choice of different treatment methods: (a) the time of bleeding and pseudoaneurysm; (b) comorbidities and complications, such as inflammation and infection; and (c) hemodynamic [18].

With respect to the treatment of arterial PSAs, surgery and endovascular techniques are the two primary options. Surgical management of PSAs includes resection with a bypass procedure, arterial ligation, and partial or complete organ removal (splenectomy). Surgical treatment is associated with increased morbidity and mortality, ranging from 10% to 50% of cases, as compared with minimally invasive treatment options [16]. The complications associated with surgery include bleeding, infection, lymphocele formation, radiculopathy, perioperative myocardial infarction, and death [58,59].

In the past, DSA has long been the gold standard for the detection of a visceral artery PSA in patients with clinical suspect of hemorrhage; high temporal and spatial resolution related to this technique should allow an optimal analysis of arterial vascular structure. Nowadays, CTA represent the gold standard technique (shorter time acquisition and high spatial resolution) in an emergency setting for the diagnosis and treatment planning of splenic artery PSAs while angiography only has a therapeutic role due to the presence of higher procedural and biological risk if compared to other non-invasive techniques, and its cost [10,11,12,13,14,15,16,17,18,19,20,21,22,23,24,25,26,27,28,29,30,31,32,33,34,35,36,37,38]. Percutaneous angioembolization efficacy ranges from 79% to 100%, with contemporary mortality rates of approximately 10–20%. The technical success rate of our study with endovascular embolization was 87.9%, and the clinical success was 82.8% [51,52,53,54,55,56,57,58,59,60].

Endovascular management of splenic artery PSAs includes different options; the selection of the best strategy depends on the involved visceral artery, PSA characteristics, the clinical scenario, and the operator experience. Transcatheter embolization with different embolic agents and devices is largely used worldwide; coil embolization or covered stent implantation are the most widespread embolization techniques [6]. Coil embolization may sacrifice a vessel with the potential risk of end-organ infarct, the covered stenting procedure preserves the vessel patency with aneurysm exclusion, and with no risk of distal ischemia. As the spleen has an extensive arterial network of collaterals, endovascular embolization with vessel sacrifice is the most used embolization technique; indeed, the spleen maintains flow via the short gastric arteries, gastroduodenal arteries, pancreatic arteries, and left gastroepiploic arteries. The use of liquid embolic agent is not recommended for the presence of extensive anastomosis with digestive arteries and the risk related of non-target embolization [61,62].

In this review, we considered mainly the PSAs of main splenic artery; for this reason, the embolization techniques most used are the isolation technique, the sandwich technique, the sac packing technique and the proximal embolization technique. The isolation technique is the most used and consists in embolization of the distal and proximal side of the parent artery for isolating the PSA; the sandwich technique consists of sac embolization and the inflow and the outflow arteries. The embolization of outflow allows us to prevent retrograde flow from collateral arterial network. The sac packing technique consists in only sac embolization, maintaining the patency of the parent artery; however, since it is very difficult to obtain a complete sac embolization, this technique needs a careful follow-up due to the risk of recanalization being very high. The proximal embolization consists in embolization only of the splenic artery segment proximal to PSA; generally, this technique is used only in case where the vasoconstriction is very high, and it could not allow distal catheterization [18,19,20]. The embolization procedure demonstrates a 96% success rate with low complication rates and high survival rate [1].

Endovascular repair using a covered stent allows for the exclusion of the aneurysm, preserving the flow through the splenic artery, thus reducing the potential risk of target organ ischemia; however, it is usually not the first choice due to the risk involved and the tortuosity of the visceral arteries, in addition to the vasoconstriction of the parent artery generally caused by the use of vasoconstrictive drugs. A recent large retrospective series covered that the stenting procedure is feasible in only 30% of cases with a high patency rate at the midterm follow-up, suggesting a transaxillary approach to improve these rates of technical success [63,64].

## 6. Conclusions

Main splenic artery PSAs are often found in emergent situation; therefore, a quick clinical overview and a prompt diagnosis may help to reduce high-rate mortality related to this clinical condition. CTA represents the best tool for diagnosis and treatment planning of splenic artery PSAs; shorter time acquisition and high spatial resolution (<1 mm) makes CTA the gold standard technique in emergency setting. Endovascular treatment with low complication rates and a high survival rate is a viable alternative to the surgical management of main splenic artery PSAs.

## Figures and Tables

**Figure 1 diagnostics-12-01012-f001:**
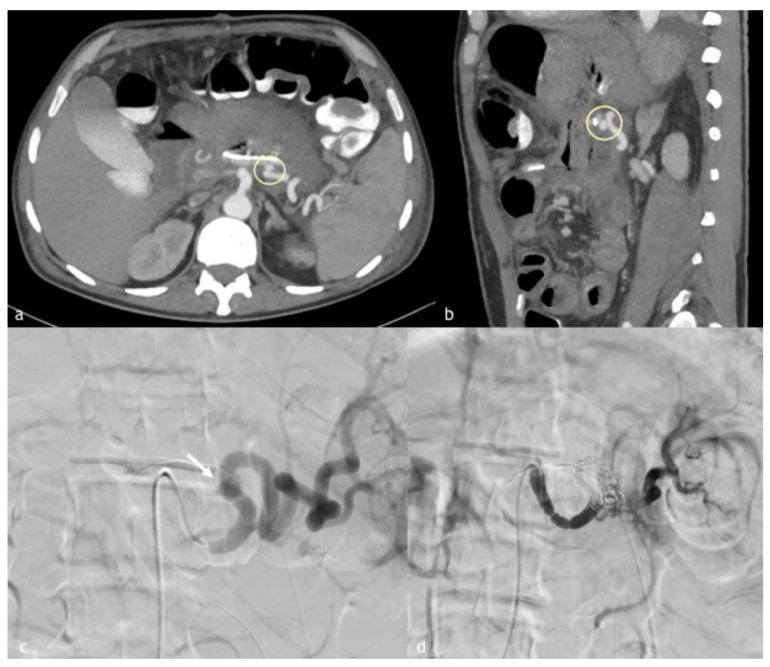
(**a**–**d**). A 60-year-old man with a walled off necrosis after necrotizing pancreatitis. (**a**,**b**) Axial and sagittal MPR reconstruction images demonstrate splenic artery PSA (circle) in direct connection with a walled-off necrosis, previously treated surgically. (**c**) Angiographic images of splenic artery PSA (arrow in (**c**)). (**d**) Final angiographic control after coil embolization demonstrates splenic artery recanalization distal to coil embolization by magna pancreatic artery.

**Figure 2 diagnostics-12-01012-f002:**
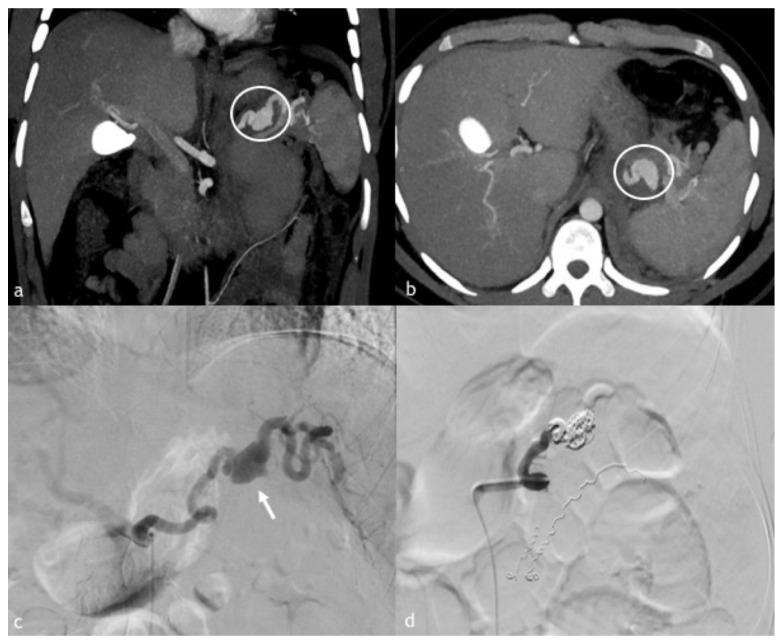
(**a**–**d**). A 20-year-old man presented to our emergency department after high blunt trauma. (**a**,**b**) Axial and coronal MPR reconstruction images demonstrate splenic artery PSA, rounded by coarse hematoma (circle). (**c**) Digital subtraction angiography of splenic artery demonstrates PSA (arrow) with the presence of multiple wall irregularities, such as blebs (a warning sign of an impending breakout). (**d**) Post-embolization angiographic control with complete embolization of the splenic artery. There are also other coils due to another adrenal hemorrhage, successfully treated in the same session.

**Figure 3 diagnostics-12-01012-f003:**
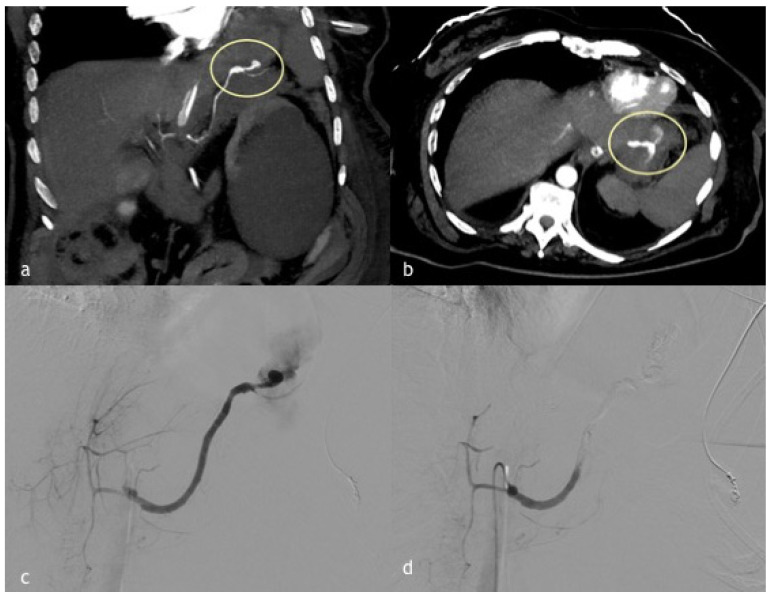
(**a**–**d**). A 45-year-old woman presented a massive hematemesis after gastroentero-anastomosis due to ingestion of caustics. (**a**,**b**) Coronal and axial MPR reconstruction images demonstrate a spastic splenic artery with PSA formation directly bleeding in the stomach (circle). (**c**) Digital subtraction angiographic image shows splenic artery PSA rupture with massive bleeding in the stomach. (**d**) Post-embolization angiographic control shows complete occlusion of the splenic artery obtained with 1:1 mixture Glue/Lipiodol.

**Figure 4 diagnostics-12-01012-f004:**
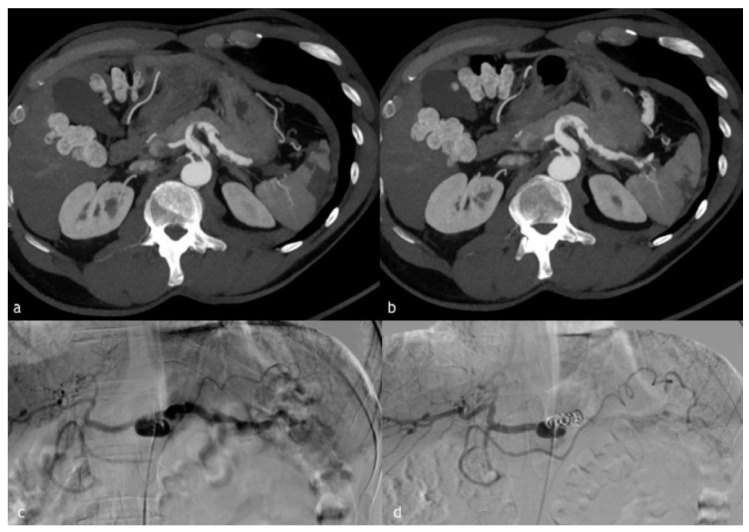
(**a**–**d**). A 55-year-old man presented to our emergency department for acute abdominal pain. (**a**,**b**) Axial MIP reconstruction images demonstrate celiac trunk dissection with multiple beading or webs, aspects of the splenic artery suggestive of SAM. (**c**) Digital subtraction angiographic image that shows multiple beading and webs of the splenic artery vessel wall. (**d**) Post-embolization angiographic control, after very proximal embolization for impossibility to more distal catheterization, demonstrating distal recanalization of splenic artery by gastroduodenal artery.

## Data Availability

Not applicable.

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
