# Peer review of "Splenic Artery Pseudoaneurysms: The Role of ce-CT for Diagnosis and Treatment Planning"

_diagnostics, 2022, doi:10.3390/diagnostics12041012_

Round 1

Reviewer 1 Report

This review manuscript was written comprehensively and should be interesting to the readers. However, I suggest that this manuscript to be revised by an English native personnel before further consideration of acceptance.

Author Response

Dear Editor,

first of all, we would like to thank you and reviewers for your time invested to provide comments to our manuscript.

We have read carefully reviewers’ observations and we have answered point by point.

Please find in the pages below our answers.

We hope we have improved the manuscript according to their revisions.

We really care about the possibility to publish our experience on your journal and we hope “Diagnostics” readers could appreciate our work.

Our bests,

The Authors

Response to Reviewer #1

Observation 1:

This review manuscript was written comprehensively and should be interesting to the readers. However, I suggest that this manuscript to be revised by an English native personnel before further consideration of acceptance.

Answer 1:

We completely agree with this observation; therefore, the manuscript has been revised by our mother tongue colleague. All manuscript’ changes are tracked in bold type.

Reviewer 2 Report

A very interesting manuscript.

The manuscript is a narrative review about main splenic artery PSAs diagnosis and endovascular treatment. 

It is very interesting as it reports pros and cons about the imaging methods.

Not many previous manuscripts report about main splenic artery pseudoaneurysm. Therefore, it is an innovative manuscript that draws the attention of the readers.

I have no comments or suggestion as the manuscript is really well written.

Thank you

Author Response

Response to Reviewer #2

Observation 1:

A very interesting manuscript.

The manuscript is a narrative review about main splenic artery PSAs diagnosis and endovascular treatment. 

It is very interesting as it reports pros and cons about the imaging methods.

Not many previous manuscripts report about main splenic artery pseudoaneurysm. Therefore, it is an innovative manuscript that draws the attention of the readers.

I have no comments or suggestion as the manuscript is really well written.

Thank you

Answer 1:

Thank you for your time invested to provide comments to our manuscript. We agree with reviewers' observations and we really care about the possibility to publish our experience on your journal and we hope “Diagnostics” readers could appreciate our work.

Round 2

Reviewer 1 Report

The English language and style were revised significantly.

Author Response

Dear Editor,

first of all, we would like to thank you and reviewers for your time invested to provide comments to our manuscript.

We have read carefully reviewers’ observations and we have answered point by point.

Please find in the pages below our answers.

We hope we have improved the manuscript according to their revisions.

We really care about the possibility to publish our experience on your journal and we hope “Diagnostics” readers could appreciate our work.

Response to Reviewer #1

Observation 1:

The English language and style were revised significantly.

Answer 1:

Thank you very much. Our bests